# LEARNING TO TREAT SEPSIS WITH MULTI-OUTPUT GAUSSIAN PROCESS DEEP RECURRENT Q-NETWORKS

## ABSTRACT

Sepsis is a life-threatening complication from infection and a leading cause of mortality in hospitals. While early detection of sepsis improves patient outcomes, there is little consensus on exact treatment guidelines, and treating septic patients remains an open problem. In this work we present a new deep reinforcement learning method that we use to learn optimal personalized treatment policies for septic patients. We model patient continuous-valued physiological time series using multi-output Gaussian processes, a probabilistic model that easily handles missing values and irregularly spaced observation times while maintaining estimates of uncertainty. The Gaussian process is directly tied to a deep recurrent Q-network that learns clinically interpretable treatment policies, and both models are learned together end-to-end. We evaluate our approach on a heterogeneous dataset of septic spanning 15 months from our university health system, and find that our learned policy could reduce patient mortality by as much as 8.2% from an overall baseline mortality rate of 13.3%. Our algorithm could be used to make treatment recommendations to physicians as part of a decision support tool, and the framework readily applies to other reinforcement learning problems that rely on sparsely sampled and frequently missing multivariate time series data.

## 1 INTRODUCTION

Sepsis is a poorly understood complication arising from infection, and is both a leading cause in patient mortality (Epstein et al. (2016)) and in associated healthcare costs (Torio & Moore (2016)). Early detection is imperative, as earlier treatment is associated with better outcomes (Seymour et al. (2017), Kumar et al. (2006)). However, even among patients with recognized sepsis, there is no standard consensus on the best treatment. There is a pressing need for personalized treatment strategies tailored to the unique physiology of individual patients. Guidelines on sepsis treatment previously centered on early goal directed therapy (EGDT) and more recently have focused on sepsis care bundles, but none of these approaches are individualized.

Before the landmark publication on the use of early goal directed therapy (Rivers et al. (2001)), there was no standard management for severe sepsis and septic shock. EGDT consists of early identification of high-risk patients, appropriate cultures, infection source control, antibiotics administration, and hemodynamic optimization. The study compared a 6-hour protocol of EGDT promoting use of central venous catheterization to guide administration of fluids, vasopressors, inotropes, and packed red-blood cell transfusions, and was found to significantly lower mortality. Following the initial trial, EGDT became the cornerstone of the sepsis resuscitation bundle for the Surviving Sepsis Campaign (SCC) and the Centers for Medicare and Medicaid Services (CMS) (Dellinger et al. (2013)).

Despite the promising results of EGDT, concerns arose. External validity outside the single center study was unclear, it required significant resources for implementation, and the elements needed to achieve pre-specified hemodynamic targets held potential risks. Between 2014–2017, a trio of trials reported an all-time low sepsis mortality, and questioned the continued need for all elements of EGDT for patients with severe and septic shock (ProCESS et al. (2014), ARISE & Group (2014), PRISM (2017)). The trial authors concluded EGDT did not improve patient survival compared to usual care but was associated with increased ICU admissions (Angus et al. (2015)). As a result, they did not recommend it be included in the updated SCC guidelines (Rhodes et al. (2017)).

Although the SSC guidelines provide an overarching framework for sepsis treatment, there is renewed interest in targeting treatment and disassembling the bundle (Lewis (2010)). A recent meta-analysis evaluated 12 randomized trials and 31 observational studies and found that time to first an-

tibiotics explained 96-99% of the survival benefit (Kalil et al. (2017)). Likewise, a study of 50,000 patients across the state of New York found mortality benefit for early antibiotic administration, but not intravenous fluids (Seymour et al. (2017)). Beyond narrowing the bundle, there is emerging evidence that a patient's baseline risk plays an important role in response to treatment, as survival benefit was significantly reduced for patients with more severe disease (Kalil et al. (2017)).

Taken together, the poor performance of EGDT compared to standard-of-care and improved understanding of individual treatment effects calls for re-envisioning sepsis treatment recommendations. Though general consensus in critical care is that the individual elements of the sepsis bundle are typically useful, it is unclear exactly when each element should be administered and in what quantity.

In this paper, we aim to directly address this problem using deep reinforcement learning. We develop a novel framework for applying deep reinforcement learning to clinical data, and use it to learn optimal treatments for sepsis. With the widespread adoption of Electronic Health Records, hospitals are already automatically collecting the relevant data required to learn such models. However, real-world operational healthcare data present many unique challenges and motivate the need for methodologies designed with their structure in mind. In particular, clinical time series are typically irregularly sampled and exhibit large degrees of missing values that are often informatively missing, necessitating careful modeling. The high degree of heterogeneity presents an additional difficulty, as patients with similar symptoms may respond very differently to treatments due to unmeasured sources of variation. Alignment of patient time series can also be a potential issue, as patients admitted to the hospital may have very different unknown clinical states and can develop sepsis at any time throughout their stay (with many already septic upon admission).

Part of the novelty in our approach hinges on the use of a Multi-output Gaussian process (MGP) as a preprocessing step that is jointly learned with the reinforcement learning model. We use an MGP to interpolate and to impute missing physiological time series values used by the downstream reinforcement learning algorithm, while importantly maintaining uncertainty about the clinical state. The MGP hyperparameters are learned end-to-end during training of the reinforcement learning model by optimizing an expectation of the standard Q-learning loss. Additionally, the MGP allows for estimation of uncertainty in the learned Q-values. For the model architecture we use a deep recurrent Q-network, in order to account for the potential for non-Markovian dynamics and allow the model to have memory of past states and actions. In our experiments utilizing EHR data from septic patients spanning 15 months from our university health system, we found that both the use of the MGP and the deep recurrent Q-network offered improved performance over simpler approaches.

## 2 BACKGROUND

In this section we outline important background that motivates our improvements on prior work.

### 2.1 DEEP Q-LEARNING

Reinforcement learning (RL) considers learning policies for agents interacting with unknown environments, and are typically formulated as a Markov decision process (MDP) (Sutton & Barto (1998)). At each time $t$, an agent observes the state of the environment, $s_t \in \mathcal{S}$, takes an action $a_t \in \mathcal{A}$, and receives a reward $r_t \in \mathbb{R}$, at which time the environment transitions to a new state $s_{t+1}$. The state space $\mathcal{S}$ and action space $\mathcal{A}$ may be continuous or discrete. The goal of an RL agent is to select actions in order to maximize its return, or expected discounted future reward, defined as $R_t = \sum_{t'=t}^{T} \gamma^{t'-t} r_{t'}$, where $\gamma$ captures tradeoff between immediate and future rewards.

Q-Learning (Watkins & Dayan (1992)) is a model-free off-policy algorithm for estimating the expected return from executing an action in a given state. The optimal action value function is the maximum discounted expected reward obtained by executing action $a$ in state $s$ and acting optimally afterwards, defined as $Q^*(s, a) = \max_\pi \mathbb{E}[R_t | s_t = s, a_t = a, \pi]$, where $\pi$ is a policy that maps states to actions. Given $Q^*$, an optimal policy is to act by selecting $\text{argmax}_a Q^*(s, a)$. In Q-learning, the Bellman equation is used to iteratively update the current estimate of the optimal action value function according to $Q(s, a) \doteq Q(s, a) + \alpha(r + \gamma \max_a Q(s', a') - Q(s, a))$, adjusting towards the observed reward plus the maximal Q-value at the next state $s'$. In Deep Q-learning a deep neural network is used to approximate Q-values (Mnih et al. (2015)), overcoming the issue that there may be infinitely many states if the state space is continuous. Denoting the parameters of the

neural network by $\theta$, Q-values $Q(s, a|\theta)$ are now estimated by performing a forward pass through the network. Updates to the parameters are obtained by minimizing a differentiable loss function, $L(s, a|\theta_i) = (r + \gamma\max_{a'}Q(s', a'|\theta_i) - Q(s, a|\theta_i))^2$, and training is usually accomplished with stochastic gradient descent.

## 2.2 PARTIAL OBSERVABILITY AND DEEP RECURRENT Q-NETWORKS

A fundamental limiting assumption of Markov decision processes is the Markov property, which is rarely satisfied in real-world problems. In medical applications such as our problem of learning optimal sepsis treatments, it is unlikely that a patient's full clinical state will be measured. A Partially Observable Markov Decision Process (POMDP) better captures the dynamics of these types of real-world environments. An extension of an MDP, a POMDP assumes that an agent does not receive the true state of the system, instead receiving only observations $o \in \Omega$ generated from the underlying system state according to some unknown observation model $o \sim \mathcal{O}(s)$. Deep Q-learning has no reliable way to learn the underlying state of the POMDP, as in general $Q(o, a|\theta) \neq Q(s, a|\theta)$, and will only perform well if the observations well reflect the underlying state. Returning to our medical application, the system state might be the patient's unknown clinical status or disease severity, and our observations in the form of vitals or laboratory measurements offer some insight into the state.

The Deep Recurrent Q-Network (DRQN) (Hausknecht & Stone (2015)) extends vanilla Deep Q-networks (DQN) by using recurrent LSTM layers (Hochreiter & Schmidhuber (1997)), which are well known to capture long-term dependencies. LSTM recurrent neural network (RNN) models have frequently been used in past applications to medical time series, such as Lipton et al. (2016). In our experiments we investigate the effect of replacing fully connected neural network layers with LSTM layers in our Q-network architecture in order to test how realistic the Markov assumption is in our application.

## 2.3 MGPS: MULTI-OUTPUT GAUSSIAN PROCESSES

Multi-output Gaussian processes (MGPs) are commonly used probabilistic models for irregularly sampled multivariate time series, as they seamlessly handle variable spacing, differing numbers of observations per series, and missing values. In addition, they maintain estimates of uncertainty about the state of the series. MGPs have been frequently applied to model patient physiological time series, e.g. Ghassemi et al. (2015), Durichen et al. (2015), Cheng et al. (2017).

Given $M$ time series (physiological labs/vitals), an MGP is specified by a mean function for each series $\{\mu_m(t)\}_{m=1}^{M}$, commonly assumed to be zero, and a covariance function or kernel $K$. Letting $f_m(t)$ denote the latent function for series $m$ at time $t$, then $K(t, t', m, m') = \text{cov}(f_m(t), f_{m'}(t'))$. Typically the actual observations are centered on the latent functions according to some distribution, e.g. $y_m(t) \sim \mathcal{N}(f_m(t), \sigma_m^2)$ with $\{\sigma_m^2\}_{m=1}^{M}$ noise parameters. We use the linear model of coregionalization covariance function with an Ornstein-Uhlenbeck base kernels $k(t, t') = e^{-|t-t'|/l}$ to flexibly model temporal correlations in time as well as covariance structure between different physiological variables. For each patient, letting $\mathbf{t}$ denote the complete set of measurement times across all observations, the full joint kernel is $K(\mathbf{t}, \mathbf{t}') = \sum_{p=1}^{P} \mathbf{B}_p \otimes k_p(\mathbf{t}_*, \mathbf{t}_*')$, where $P$ denotes the number of mixture kernel. $\mathbf{t}_*$ denotes the time vector for each physiological sign, assumed here to be the same for notational convenience, but in practice the full kernel need only be computed at the observed variables. Each $\mathbf{B}_p \in \mathbb{R}^{M \times M}$ encodes the scale covariance between different time series. We found that $P = 2$ works well in practice and allows learning of correlations on both short and long time scales. Given the MGP kernel hyperparameters shared across all patients, collectively referred to as $\eta$, imputation and interpolation at arbitrary times can be computed either using the posterior mean or the full posterior distribution over unknown function values.

## 2.4 MGP-RNNS: MULTI-OUTPUT GAUSSIAN PROCESS RECURRENT NEURAL NETWORKS

Multi-output Gaussian processes and recurrent neural networks can be combined and trained end-to-end (MGP-RNNs), in order to solve supervised learning problems for sequential data (Futoma et al. (2017a), Futoma et al. (2017b)). This methodology was shown to exhibit superior predictive performance at early detection of sepsis from clinical time series data, when compared with vanilla RNNs with last-one-carried-forward imputation. In fitting the two models end-to-end, the MGP

hyperparameters are learned discriminatively, in essence learning an imputation and interpolation mechanism tuned for the supervised task at hand.

Learning an MGP-RNN consists of minimizing an expectation of some loss function, with respect to the posterior distribution of the MGP. Letting $\mathbf{z}$ denote a set of latent time series values distributed according to an MGP posterior, and $g(\mathbf{z})$ denote the prediction(s) made by an RNN from this time series, then the goal is to minimize $\mathbb{E}_{\mathbf{z} \sim \mathcal{MGP}}[l(\mathbf{o}, g(\mathbf{z}))]$, where $l$ is some loss function (e.g. cross-entropy for a classification task) and $\mathbf{o}$ is the true label(s). We can express the MGP distributed latent variable $\mathbf{z}$ as $\mathbf{z} = \mu_z + R_z \xi$, where $\mu_z$ is the posterior mean and $R_z R_z^\top = \Sigma_z$ with $\Sigma_z$ the posterior covariance, and $\xi \sim (0, I)$. This allows us to apply the reparameterization trick (Kingma & Welling (2014)) and use Monte Carlo sampling to compute approximate gradients of this expectation with respect to both MGP hyperparameters $\eta$ and RNN parameters $\theta$, so that the loss can be minimized via stochastic gradient descent. The stochasticity in this learning procedure introduced from the Monte Carlo sampling additionally acts as a form of regularization, and helps prevent the RNN from overfitting. In Section 3 we show how this can be applied to a reinforcement learning task.

## 2.5 RELATED WORK FROM REINFORCEMENT LEARNING IN HEALTHCARE

There has been substantial recent interest in development of machine learning methodologies motivated by healthcare data. However, most prior work in clinical machine learning focuses on supervised tasks, such as diagnosis (Esteva et al. (2017)) or risk stratification (Futoma et al. (2017a)). Many recent papers have developed models for early detection of sepsis, a related problem to our task of learning treatments for sepsis, e.g. Soleimani et al. (2017), Henry et al. (2015), Futoma et al. (2017b). However, as supervised problems rely on ground truth they cannot be applied to treatment recommendation, unless the assumption is made that past training examples of treatments represent optimal behavior. Instead, it is preferable to frame the problem using reinforcement learning in order to learn optimal treatment actions from data collected from potentially suboptimal actions.

While deep reinforcement learning has seen huge success over the past few years, only very recently have reinforcement learning methods been designed with healthcare applications in mind. Applying reinforcement learning methods to healthcare data is difficult, as it requires careful consideration to set up the problem, especially the rewards. Furthermore, it is typically not possible to collect additional data and so evaluating learned policies on retrospective data presents a challenge.

Most related to this paper are Raghu et al. (2017) and Komorowski et al. (2016), who also look at the problem of learning optimal sepsis treatments. We build off of their work by using a more sophisticated network architecture that takes into account both memory through the use of DRQNs and uncertainty in time series imputation and interpolation using MGPs. Other relevant work includes Prasad et al. (2017), who use a simpler learning algorithms to learn optimal strategies for ventilator weaning, and Nemati et al. (2016), who also use a deep RL approach for modeling ICU heparin dosing as a POMDP with discriminative hidden Markov models and Q-networks. There also exists a rich set of work from the statistics and causal inference literature on learning dynamic treatment regimes, e.g. Chakraborty & Moodie (2013), Shortreed et al. (2010), although the models are typically fairly simple for ease of interpretability.

## 3 MGP-DRQN: MULTI-OUTPUT GAUSSIAN PROCESS DEEP RECURRENT Q-NETWORKS

We now introduce Multi-Output Gaussian Process Deep Recurrent Q-Networks, or MGP-DRQNs, a novel reinforcement learning algorithm for learning optimal treatment policies from noisy, sparsely sampled, and frequently missing clinical time series data. We assume a discrete action space, $a \in \mathcal{A} = \{1, \ldots, A\}$. Let $\mathbf{x}$ denote $T$ regularly spaced grid times at which we would like to learn optimal treatment decisions. Given a set of clinical physiological time series $\mathbf{y}$ that we assume to be distributed according to an MGP, we can compute a posterior distribution for $z_t | \mathbf{y}$, the latent unobserved time series values at each grid time.

The loss function we optimize is similar to in normal deep Q-learning, with the addition of the expectation due to the MGP and the fact that we compute the loss over full patient trajectories. In

particular, we learn optimal DRQN parameters $\theta^*$ and MGP hyperparameters $\eta^*$ via:

$$\theta^*, \eta^* = \mathrm{argmin}_{\theta, \eta} \mathbb{E} \left[ \mathbb{E}_{p(\mathbf{z}|\mathbf{y};\eta)} \left\{ \frac{1}{T} \sum_{t=1}^{T} (Q_{target}^{(t)} - Q([z_t, s_t]^\top, a; \theta))^2 \right\} \right],$$

where the $t$'th target value is $Q_{target}^{(t)} = r_t + \gamma \max_{a'} Q([z_{t+1}, s_{t+1}], a')$, the outer expectation is over training samples, and the inner one is with respect to the MGP posterior for one patient. We concatenate the two separate types of model inputs at time $t$, with $z_t$ denoting latent variables distributed according to an MGP posterior from other relevant inputs to the model denoted $s_t$, such as static baseline covariates. In Section 4.1 we go into detail on the particular variables included in $s_t$.

We use a Dueling Double-Deep Q-network architecture, similar to Raghu et al. (2017). The Dueling Q-network architecture Wang et al. (2016) has separate value and advantage streams to separate the effect of a patient being in a good underlying state from a good action being taken. The Double-Deep Q-network architecture (van Hasselt et al. (2016)) helps correct overestimation of Q-values by using a second target network to compute the Q-values in the target $Q_{target}$. Finally, we use Prioritized Experience Replay in order to speed learning, so that patient encounters with higher training error will be resampled more frequently. We use 2 LSTM layers with 64 hidden units each that feed to a final fully connected layer with 64 hidden units, before splitting into equally sized value and advantage streams that are finally then projected onto the action space to obtain Q-value estimates.

We implemented our methods in Tensorflow using the Adam optimizers (Kingma & Ba (2015)) with minibatches of 50 encounters sampled at a time, a learning rate of 0.001, and $L_2$ regularization on weights. We use 25 Monte Carlo samples from the MGP for each sampled encounter in order to approximate the expected loss and compute approximate gradients, and these samples and other inputs are fed in a forward pass through the DRQN to get predictions $Q(s, a)$. We will release source code via Github after the review period.

## 4 EXPERIMENTS, EVALUATION, AND RESULTS

In this section we first describe the details of our dataset of septic patients before highlighting how the experiments were set up and how the algorithms were evaluated.

### 4.1 DATASET AND PREPROCESSING

Our dataset consists of information collected during 9,255 patient encounters resulting in sepsis at our university hospital, spanning a period of 15 months. We define sepsis to be the first time at which a patient simultaneously had persistently abnormal vitals (as measured by a 2+ SIRS score, Bone et al. (1992)), a suspicion of infection (as measured by an order for a blood culture), and an abnormal laboratory value indicative of organ damage. This differs from the new Sepsis-3 definition (Seymour et al. (2016)), which has since been largely criticized for its detection of sepsis late in the clinical course (Cortes-Puch & Hartog (2016)). We break the full dataset into 7867 training patient encounters and reserve the remaining 1388 for testing.

We discretize the data to learn actions in 4 hour windows. We emphasize that the raw data itself is not down-sampled; rather, we use the MGP to learn a posterior for the time series values every 4 hours. Actions for the RL setup consist of 3 treatments commonly given to septic patients: antibiotics, vasopressors, and IV fluids. Antibiotics and vasopressors are broken down into 3 categories, based on whether 0, 1, or 2+ were administered in each 4 hour window. For IV Fluids, we consider 5 discrete categories: either 0, or one of 4 aggregate doses based on empirical quartiles of total fluid volumes. This yields a discrete action space with $3 \times 3 \times 5 = 45$ distinct actions.

Our data consists of 36 longitudinal physiological variables (e.g. blood pressure, pulse, white blood cell count), 2 longitudinal categorical variables, and 38 variables available at baseline (e.g. age, previous medical conditions). 8 medications tangential to sepsis treatment are included as inputs to MGP-DRQN, as well as an indicator for which of the 45 actions was administered at the last time. Additionally, 36 indicator variables for whether or not each lab/vital was recently sampled allows the model to learn from informative sampling due to non-random missingness. In total, there are 165 input observation variables to each of the Q-network models at each time.

Our outcome of interest is mortality within 30 days of onset of sepsis. We use a sparse reward function in this initial work, so that the reward at every non-terminal time point is 0, with a reward of $\pm 10$ at the end of a trajectory based on patient survival/death. Although this presents a challenging credit assignment problem, this allows for data to inform what actions should be taken to reduce chance of death without being overly prescriptive.

## 4.2 BASELINE METHODS

We use SARSA, an on-policy algorithm, to estimate state-action values for the physician policy.

We compare a number of different architectures for learning optimal sepsis treatments. In addition to our proposed MGP-DRQN, we compare against MGP-mean-DRQN, a variant where we move the posterior expectation inside the DRQN loss function, meaning we use the posterior mean of the MGP rather than use Monte Carlo samples from the MGP. We also compare against a DRQN with identical architecture, but replace the MGP with last-one-carried-forward imputation to fill in any missing values, and use the mean if there are multiple measurements. We also compare against a vanilla DQN, a MGP-DQN, and a MGP-mean-DQN, with an equivalent number of layers and parameters, to test the effect of the recurrence in the DRQN models.

## 4.3 OFF-POLICY VALUE EVALUATION

We use Doubly Robust Off-policy Value Evaluation (Jiang & Li (2016)) to compute unbiased estimates of each learned optimal policy using our observed off-policy data. For each patient trajectory in the test set we estimate its value using this method, and the average results. In order to apply this method we train an MGP-RNN to estimate the action probabilities of the physician policy.

## 4.4 QUANTITATIVE RESULTS

In Figure 1 we show the results of using SARSA to estimate expected returns for the physician policy on the test data. The Q-values appear to be well calibrated with mortality, as patients who were estimated to have higher expected returns tended to have lower mortality. Due to small sample sizes for very low expected returns, the mortality rate does not always monotonically decrease.

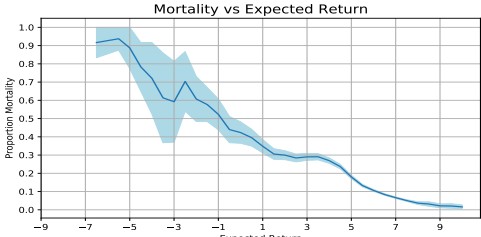

Figure 1: For the 1388 patients in the test set we show the expected returns as computed by SARSA, against 30-day mortality among patients with similar Q-values. Our model appears to be well calibrated, as higher returns are associated with lower mortality.

We can estimate the potential reduction in mortality a learned policy might have by computing an unbiased estimate of the policy value, as described in Section 4.3, and then use the results in Figure 1. Table 1 contains the policy value estimates for each algorithm considered, along with estimated mortality rates. The physician policy has an estimated value of 5.52 and corresponding mortality of 13.3%, matching the observed mortality in the test set of 13.3%. Overall the MGP-DRQN performs and might reduce mortality by as much as 8%. The DRQN architectures tended to yield higher expected returns, probably because they are able to retain some memory of past clinical states and actions taken. The MGP consistently improved results as well, and the additional uncertainty information contained in the full MGP posterior appeared to do better than the policies that only used the posterior mean.

| Policy | Expected Return | Estimated Mortality |
|---|---|---|
| Physician | 5.52 | 13.3 ± 0.7% |
| **MGP-DRQN** | **7.51** | **5.1 ± 0.5%** |
| MGP-mean-DRQN | 6.97 | 6.6 ± 0.4% |
| DRQN | 6.63 | 8.4 ± 0.4% |
| MGP-DQN | 7.05 | 6.6 ± 0.4% |
| MGP-mean-DQN | 6.73 | 7.5 ± 0.4% |
| DQN | 6.09 | 10.6 ± 0.5% |

Table 1: Expected returns for the various policies considered. For the 6 reinforcement learning algorithms considered, we estimate their expected returns using an off-policy value evaluation algorithm. Using the results from Figure 1, we estimate the potential expected mortality reduction associated with each policy.

### 4.5 QUALITATIVE RESULTS

We also qualitatively evaluate the results of the policy from our best performing learning algorithm, the MGP-DRQN. In Figure 2 we compare the number of times each type of action was actually taken by physicians, and how many times the learned policy selected that action. The MGP-DRQN policy tended to recommend more use of antibiotics and more vasopressors than were actually used by physician, while strangely recommending somewhat less use of IV fluids.

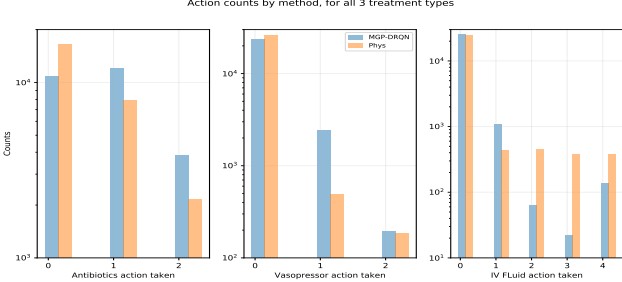

Figure 2: Comparison of physician actions with the actions that would have been taken by the MGP-DRQN policy, with actions separated according to the 3 types of treatments considered.

In Figure 3, we show how mortality rates differ on the test set as a function of how different the observed physician action was from what the MGP-DRQN would have recommended. For all 3 types of treatments, there appears to be a local minimum at 0 and we observe a V shape, indicating that empirically, mortality tended to be lowest when the clinicians took the same actions that the MGP-DRQN would have. Uncertainty tends to be higher due to smaller sample sizes for situations where there is larger disparity.

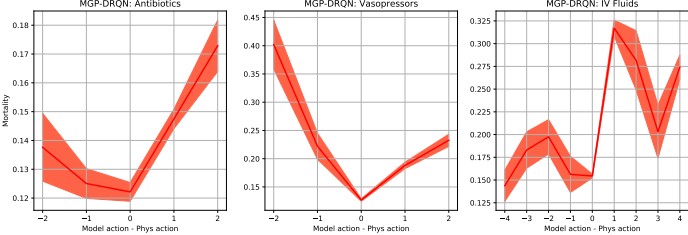

Figure 3: Empirical mortality rates as a function of how much the MGP-DRQN policy's actions differed from the observed physician actions. Minimal mortality is observed for all 3 treatment types at 0, where the physicians and MGP-DRQN agreed.

Finally, in Figure 4 we show clinical data from a sample patient case. In the top pane of the figure we show five representative vital signs and lab measurements to illustrate the patient's clinical status, while the bottom shows both what actions physicians actually took and what actions the model

recommended. The patient was admitted to the Emergency Department for altered mental status, and the MGP-DRQN quickly recognizes the need for antibiotics and IV fluids. The patient is admitted to the hospital and around hour 6 the clinical team becomes aware of sepsis. However, antibiotics are not first administered until hour 18, about 16 hours after the model recommended treating with them. After the patient is transferred to the Intensive Care Unit, their white blood cell count continues to rise (a sign of worsening infection) and their blood pressure continues to fall (a sign of worsening shock). By hour 14, the RL model starts and continues to recommend use of vasopressors to attempt to increase blood pressure, but they are not actually administered for about another 16 hours at hour 30. Ultimately, by hour 45 care was withdrawn and the patient passed away at hour 50. Cases such as this one illustrate the potential benefits of using our learned treatment policy in a decision support tool to recommend treatments to providers. If such a tool were used in this situation, it is possible that earlier treatments and more aggressive interventions might have resulted in a different outcome.

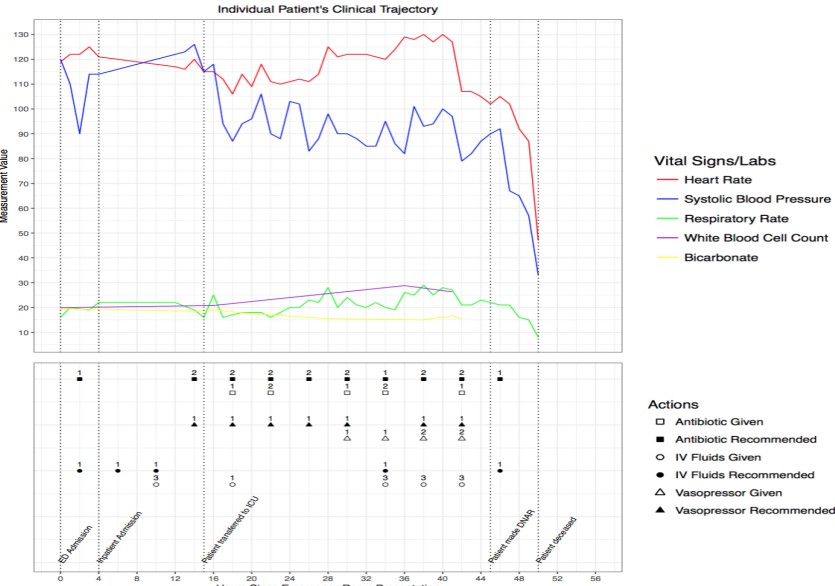

Figure 4: Top: clinical data from a patient who acquired sepsis, decompensated in the Intensive Care Unit while progressing to septic shock, and ultimately did not survive. Bottom: shaded symbols denote treatments that the learned MGP-DRQN policy would have recommended, while open symbols denote the treatment actions actually taken by physicians caring for this patient.

## 5 CONCLUSION

In this paper we presented a new framework combining multi-output Gaussian processes and deep reinforcement learning for clinical problems, and found that our approach performed well in estimating optimal treatment strategies for septic patients. The use of recurrent structure in the Q-network architecture yielded higher expected returns than a standard Q-network, accounting for the non-Markovian nature of real-world medical data. The multi-output Gaussian process also improved performance by offering a more principled method for interpolation and imputation, and use of the full MGP posterior improved upon the results from just using the posterior mean.

In the future, we could include treatment recommendations from our learned policies into our dashboard application we have developed for early detection of sepsis. The treatment recommendations might help providers better care for septic patients after sepsis has been properly identified, and start treatments faster. There are many potential avenues for future work. One promising direction is to investigate the use of more complex reward functions, rather than the sparse rewards used in this work. More sophisticated rewards might take into account clinical targets for maintaining hemodynamic stability, and penalize an overzealous model that recommends too many unnecessary actions. Our modeling framework is fairly generalizable, and can easily be applied to other medical applications where there is a need for data-driven decision support tools. In future work we plan to use similar methods to learn optimal treatment strategies for treating patients with cardiogenic shock, and to learn effective insulin dosing regimes for patients on high-dose steroids.

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
