# OpenReview forum: "Learning to Treat Sepsis with Multi-Output Gaussian Process Deep Recurrent Q-Networks"
_ICLR.cc/2018/Conference — Reject_

### Official Review · AnonReviewer3 · 2017-11-27
**Important application, but technical novelty unclear**

**Rating:** 3
**Confidence:** 4

**Review:**

The paper presents an application of deep learning to predict optimal treatment of sepsis, using data routinely collected in a hospital. The paper is very clear and well written, with a thorough review of related work. However, the approach is mainly an application of existing methods and the technical novelty is low. Further, the methods are applied to only a single dataset and there is no comparison against the state of the art, only between components of the method. This makes it difficult to assess how much of an improvement this collection of methods provides and how much it would generalize to data from other hospitals or applications. As written, the paper may be more appropriate for an application-focused venue.

---

> ### Author Response · Authors · 2018-01-02
> **Thank you for the comments and feedback, but we disagree about novelty and venue**
>
> Thank you for the comments and feedback.
>
> However, we strongly disagree about the novelty of our work and the overall contribution. Although the constituent methods we rely on in this work are not novel, their combination together in this setting is novel. As noted by R1, an important takeaway from our work is confirmation and replication of existing work, as we show that we can use RL to improve upon current clinical practice in treatment of sepsis.
>
> It is a valid criticism that we only applied our methods to a single dataset.  In the future we plan to apply our methods to the more readily available MIMIC data as well, as a second dataset. However, it takes an enormous amount of manual effort to clean and prepare raw clinical data from an electronic health record for analysis. Even the MIMIC data requires a lot of preprocessing before modeling.  In our particular application of interest at an academic university hospital, MIMIC data is not that useful because it only contains data from an ICU setting, and we are interesting in treating sepsis in other clinical settings as well (in our data, only about 20% of sepsis cases first present in the ICU).
>
> The DQN baseline method we compare against is extremely similar to Raghu et al, which to our knowledge is state of the art.  We will make this distinction more clear. There are not many papers published that apply modern reinforcement learning methods to observational clinical data, to our knowledge.  If the reviewer has a specific state of the art method that we should compare against, we would be happy to compare against it and add it to our results. We believe the ablation study we performed examining the utility of the MGP and the recurrent architecture is convincing, as (see R1 comment) all our baseline methods shared the same architecture setup and hyperparameters.
>
> We strongly disagree that it is difficult to assess what improvement these methods provide; our results show clear improvements to using an MGP for interpolation and data augmentation, and recurrence to learn the latent state. It is true that the particular learned policy may not generalize well if it were to be applied to very different patient populations, but the overall method would still apply and a new policy could be learned from that population instead.  In future work, we could compare how the policy learned on our institutional data might perform on MIMIC (but there is no reason to suspect it would be great, as MIMIC is only ICU patients, which is an extremely different population).  As R1 notes, it is an important finding in and of itself that we can use RL and it seems to work well in solving a real clinical problem, confirming the previous findings of Raghu et al.
>
> We believe that this work is appropriate for ICLR, as one of the relevant topics on the conference website is applications, which our work would clearly fall under.

---

### Official Review · AnonReviewer2 · 2017-11-27
**Re: Learning to Treat Sepsis with Multi-Output Gaussian Process Deep Recurrent Q-Networks**

**Rating:** 4
**Confidence:** 4

**Review:**

The paper presents a reinforcement learning method that uses Q-learning with deep neural networks and a multi-output Gaussian process for imputation, all for retrospective analysis of treatment decisions for preventing mortality among patient with sepsis.

While the work represents a combination of leading methods in the machine learning literature, key details are missing: most importantly, that the reinforcement learning is based on observational data and in a setting where the unconfoundedness assumption is very unlikely to hold. For example, an MGP imputation implicitly assumes MAR (missing at random) unless otherwise specified, e.g. through informative priors. The data is almost certainly MNAR (missing not at random). These concerns ought to be discussed at length.

The clarity of the work would be improved with figures describing the model (e.g. plate/architecture diagram) and pseudocode. E.g. as it stands, it is not clear how the doubly-robust estimation is being used and if it is appropriate given the above concerns. Similar questions for Dueling Double-Deep Q-network, Prioritized Experience Replay.

The medical motivation does frame the clinical problem well. The paper does serve as a way to generate hypotheses, e.g. greater use of abx and vasopressors but less IVF.

The results in Table 1 suggest that the algorithmic policy would prevent the death of ~1 in 12 individuals (ARR 8.2%) that a physician takes care of in your population. The text says "might reduce mortality by as much as 8%". The authors might consider expanding on this. What can/should be done convince the reader this number is real.

Additional questions: what is the sensitivity of the analysis to time interval and granularity of the action space (here, 4 hours; 3x3x5 treatments)? How would this work for whole order sets? In the example, abx 1 and abx 2 are recommended in the next 4 hours even after they were already administered. How does this relate to pharmacologic practice, where abx are often dosed at specific, wider intervals, e.g. vancomycin q12h? How could the model be updated for a clinician who acknowledges the action suggestion but dismisses it as incorrect?

---

> ### Author Response · Authors · 2018-01-02
> **Thank you for the insightful comments and constructive feedback**
>
> Thank you for the insightful comments and constructive feedback. We are revising the paper to address your concerns, and the revision will be posted shortly.
>
> - These concerns about observational data and unconfoundedness are fair, and we will make these assumptions more explicit.  Although the MGP makes a MAR assumption, in our models we explicitly model the missingness structure of the time series data by using indicator variables representing whether or not a particular variable was recently sampled. Empirically in past work we found modeling this missing data structure to be very helpful, and we will make it more clear that we are doing this. We do not feel that the underlying MAR assumption that comes with the MGP is overly restrictive, as the MGP primarily functions as a preprocessing step to do better interpolation and function as a form of  data augmentation to reduce overfitting.  As for the unconfoundedness assumption, this is an extremely common assumption in causal inference and off-policy reinforcement learning; we will make this more explicit.
> - We will add a schematic diagram detailing the model architecture and how the MGP feeds into the downstream DRQN.  As discussed by R1, we have updated our discussion on off-policy evaluation. In all baseline methods we used dueling double-deep Q-networks and prioritized experience replay, as in Raghu et al. We will make these modifications more clear.
> - As noted by R1, there is some discussion warranted regarding the use of off-policy evaluation, so these concerns are valid. Thus our final mortality reduction estimates may be somewhat optimistic. Rather than dwelling on exact quantities, the take-home message is that the proposed architectures seem to offer improvements, and it seems that simply using an RL approach can improve over current physician policy.
> - We did not do an extensive sensitivity analysis of the time interval size and granularity of the action space, but in our experience changing these did not seem to greatly impact performance. We chose a fairly long (4 hour) time interval, in order to reduce the number of times the "no treatment" action would be taken, as of course this gets more common with a finer time window. We chose the action space somewhat heuristically, aiming to make it as fine as possible, while checking to make sure that almost all of the 45 different actions occur at least a reasonable number of times.  It might be worth checking to see what the coarsest possible action space would be that does not compromise performance.
> - For whole order sets, we might want to change the action space to have each action be a different (commonly ordered) order set, rather than the way we've broken down the actions.  This would certainly be a more actionable and more directly applicable problem setup. This is an interesting idea, but for now we leave it to future work.
> - Abx 1 and Abx 2 simply refers to number of antibiotics given, not specific classes. Eg Abx 1 is simply "1 abx given in this 4 hour window", and Abx 2 is "2 or more abx given in this 4 hour window".  As with the previous comment, in future work we will aim to build a more directly actionable action space. Probably in this particular example, since WBC continued to rise, the RL model continued to recommend more abx be given, even after they were already administered. This might be reasonable, since in practice there were 5/7 4 hour windows where at least one abx was given. For dosing specific drugs in wider intervals, we'd want to increase the action space to more directly take into account timing and dosing of different drugs, eg if we want a 6h or a 12h version of a drug.
> - This is a great suggestion, and a very interesting avenue for future work. We would want a clinical decision support tool to log what clinicians actually ended up doing after viewing the RL recommended action, and also log whether or not a clinician views an action suggestion as wrong. Given a dataset of clinician actions and responses to the RL suggestions, we could retrain the model and explicitly penalize cases where the RL model made an incorrect suggestion. There is definitely room for future work here - a caveat is that we don't want to entirely discredit what the model recommends, as there is still room for physician error in judging the model suggestions, and there may be cases where the suggestion actually would have been good.

---

### Official Review · AnonReviewer1 · 2017-11-28
**Strong application paper synthesizes several existing ideas to apply RL to sepsis treatment**

**Rating:** 6
**Confidence:** 3

**Review:**

This paper presents an important application of modern deep reinforcement learning (RL) methods to learning optimal treatments for sepsis from past patient encounters. From a methods standpoint, it offers nothing new but does synthesize best practice deep RL methods with a differentiable multi-task Gaussian Process (GP) input layer. This means that the proposed architecture can directly handle irregular sampling and missing values without a separate resampling step and can be trained end-to-end to optimize reward -- patient survival -- without a separate ad hoc preprocessing step. The experiments are thorough and the results promising. Overall, strong application work, which I appreciate, but with several flaws that I'd like the authors to address, if possible, during the review period. I'm perfectly willing to raise my score at least one point if my major concerns are addressed.

QUALITY

Although the core idea is derivative, the work is executed pretty well. Pros (+) and cons (-) are listed below:

+ discussion of the sepsis application is very strong. I especially appreciated the qualitative analysis of the individual case shown in Figure 4. While only a single anecdote, it provides insight into how the model might yield clinical insights at the bedside.
+ thorough comparison of competing baselines and clear variants -- though it would be cool to apply offline policy evaluation (OPE) to some of the standard clinical approaches, e.g., EGDT, discussed in the introduction.

- "uncertainty" is one of the supposed benefits of the MTGP layer, but it was not at all clear how it was used in practice, other than -- perhaps -- as a regularizer during training, similar to data augmentation.
- uses offline policy evaluation "off-the-shelf" and does not address or speculate the potential pitfalls or dangers of doing so. See "Note on Offline Policy Evaluation" below.
- although I like the anecdote, it tells us very little about the overall policy. The authors might consider some coarse statistical analyses, similar to Figure 3 in Raghu, et al. (though I'm sure you can come up with more and better analyses!).
- there are some interesting patterns in Table 1 that the authors do not discuss, such as the fact that adding the MGP layer appears to reduce expected mortality more (on average) than adding recurrences. Why might this be (my guess is data augmentation)?

CLARITY

Paper is well-written, for the most part. I have some nitpicks about the writing, but in general, it's not a burden to read.

+ core ideas and challenges of the application are communicated clearly

- the authors did not detail how they chose their hyperparameters (number of layers, size of layers, whether to use dropout, etc.). This is critical for fully assessing the import of the empirical results.
- the text in the figures are virtually impossible to read (too small)
- the image quality in the figures is pretty bad (and some appear to be weirdly stretched or distorted)
- I prefer the X-axis labels that Raghu uses in their Figure 4 (with clinically interpretable increments) over the generic +1, +2, etc., labels used in Figure 3 here

Some nitpicks on the writing

* too much passive voice. Example: third paragraph in introduction ("Despite the promising results of EGDT, concerns arose."). Avoid passive voice whenever possible.
* page 3, sec. 2.2 doesn't flow well. You bounce back and forth between discussion of the Markov assumption and full vs. partial observability. Try to focus on one concept at a time (and the solution offered by a proposed approach). Note that RNNs do NOT relax the Markov assumption -- they simply do an end run around it by using distributed latent representations.

ORIGINALITY

This work scores relatively low in originality. It really just combines ideas from two MLHC 2017 papers [1][2]. One could read those two papers and immediately conclude this paper's findings (the GP helps; RL helps; GP + RL is the best). This paper adds few (if any) new insights.

One way to address this would be to discuss in greater detail some potential explanations for why their results are stronger than those in Raghu and why the MTGP models outperform their simpler counterparts. Perhaps they could run some experiments to measure performance as a function of the number of MC samples (if perhaps grows with the number of samples, then it suggests that maybe it's largely a data augmentation effect).

SIGNIFICANCE

This paper's primary significance is that it provides further evidence that RL could be applied successfully to clinical data and problems, in particular sepsis treatment. However, this gets undersold (unsurprising, given the ML community's disdain for replication studies). It is also noteworthy that the MTGP gives such a large boost in performance for a relatively modest data set -- this property is worth exploring further, since clinical data are often small. However, again, this gets undersold.

One recommendation I would make is that the authors directly compare the results in this paper with those in Raghu and to point out, in particular, the confirmatory results. Interestingly, the shapes of the action vs. mortality rate plots (Figure 4 in Raghu, Figure 3 here) are quite similar -- that's not precisely replication, but it's comforting.

NOTE ON OFFLINE POLICY EVALUATION

This work has the same flaw that Raghu, et al., has -- neither justifies the use of offline policy evaluation. Both simply apply Jiang, et al.'s doubly robust approach [3] "off the shelf" without commenting on its accuracy in practice or discussing potential pitfalls (neither even considers [4] which seems to be superior in practice, especially with limited data). As far as I can tell (I'm not an RL expert), the DR approach carries stronger consistency guarantees and reduced variance but is still only as good the data it is trained on, and clinical data is known to have significant bias, particularly with respect to treatment, where clinicians are often following formulaic guidelines. Can we trust the mortality estimates in Table 1? Why or why not? Why shouldn't I think that RL is basically guaranteed to outperform non-RL approaches under an evaluation that is itself an RL model learned from the same training data!

While I'm willing to accept that this is the best we can do in this setting (we can't just try the learned policy on new patients!), I think this paper (and similar works, like Raghu, et al.) *must* provide a sober and critical discussion of its results, rather than simply applaud itself for getting the best score among competing approaches.

REFERENCES

[1] Raghu, et al. "Continuous State-Space Models for Optimal Sepsis Treatment - a Deep Reinforcement Learning Approach." MLHC 2017.
[2] Futoma, et al. "An Improved Multi-Output Gaussian Process RNN with Real-Time Validation for Early Sepsis Detection." MLHC 2017.
[3] Jiang, et al. "Doubly robust off-policy value evaluation for reinforcement learning." ICML 2016.
[4] Thompson and Brunskill. "Data-Efficient Off-Policy Policy Evaluation for Reinforcement Learning." ICML 2016.

---

> ### Author Response · Authors · 2018-01-02
> **Thank you for the insightful comments and constructive feedback**
>
> Thank you for the insightful comments and constructive feedback. We are revising the paper to address your concerns, and the revision will be posted shortly.
>
> "Quality" comments:
> + We believe an important practical use for our method is in identifying treatments earlier than they were actually given, as evidenced in our single example.
> + Practically these comparisons would be difficult to directly make, since in our observational data there is no guarantee about how often standard clinical approaches such as EGDT are actually followed. Although possible in principle to define a computable treatment strategy to try to mimic, eg EGDT (if A then give X, if B then give Y, ...), in practice this would be pretty hard to define. This is a cool idea though that we'll leave to future work.
> - The uncertainty mostly acts as a regularizer, yes. It is also a form of data augmentation, since from a single set of patient clinical time series we get multiple draws from the MGP. Empirically and in past work we've found it reduces overfitting compared to using the MGP mean. It is possible to utilize the associated uncertainty as well in learning the policy, although we did not explore this much. The uncertainty in time series inputs captured by the MGP can be propagated forwards through the DRQN to the learned Q-values, giving some notion of uncertainty in Q-values due to uncertain inputs. Combining this with other Bayesian deep learning methods might give improved uncertainty quantification, which could be useful in learning an optimal (potentially stochastic) policy.
> - Will address OPE below.
> - It is hard to concisely summarize a policy - much room for future work here! Since we have 3 types of actions instead of 2 in Raghu et al it is hard to reproduce this figure, since our policy would need to be visualized as a 3d tensor not a matrix. We explored using a histogram that enumerates all 45 possible actions, but it was very cluttered. An analysis that we have added is checking how often the learned RL policy makes recommendations for treatments before they were actually given by a physician. This gives some notion of how timely a learned policy is, if in many cases it is recommending the same treatments that were eventually given, only sooner.
> - Yes, probably due to the reduction in overfitting associated with the data augmentation effect of the MGP, as during training the MGP provides many inputs by drawing samples from a single set of patient data.
>
> "Clarity" comments:
> + Thank you!
> - We address this more explicitly in the revision. We did not do much widespread experimentation with hyperparameters. We used the same neural network architecture across all methods, in terms of number of layers, layer size, learning rates, etc so  it is unlikely our observed results are due to hyperparameters, though improved performance for some methods may be possible by more careful tuning.
> - Will correct text size.
> - Will fix image resolution (we initially tried to fit everything in 8 pages)
> - Will edit these axis labels for IV. For antibiotics and vasopressors, however, our action space depends on quantity and not actual dosing: +1, +2 refers to number of times a drug in a class was given within the time window, and not dosing. Our clinical collaborators advised that this makes more sense, especially since for vasopressors the dosing is very unclear and would be hard to quantify numerically due to differences in drugs; we're not sure how Raghu et al assessed vasopressor dosing.
> * Have made some edits to the writing
> * Have revised sec 2.2 to flow better, making more explicit your correct point that we are not relaxing the Markov assumption, but instead use a latent representation that depends on the full history so far.

---

> > ### Author Response · Authors · 2018-01-02
> > **Thank you for the insightful comments and constructive feedback (cont.)**
> >
> > "Originality" comments:
> > - We feel that this combination of GP + RL is interesting and useful. While constituent pieces are not themselves novel, we emphasize that the loss function that we optimize is in fact novel.  The use of the GP acts as a form of data augmentation and extra regularization that helped empirically.
> > - This is a good idea, and we will try to expand upon this more with experiments comparing MC sample sizes.
> >
> > "Significance":
> > - These are both excellent points, and we have updated our discussion to better emphasize them.
> > - This is also a good point. It is not feasible in the short term to make a direct comparison against Raghu, et al on the same dataset.  However, our DQN baseline method is roughly equivalent to their methodology, and we will make this comparison more explicit.
> >
> > In future work, we will also run our method on MIMIC data, so we can have a more direct comparison. It is worth reiterating that our dataset, unlike MIMIC, is not constrained to only ICU patients, but includes patients in every area of the hospital, including Emergency Department, general wards, and ICU.
> >
> > "Note on Offline Policy Evaluation" comments:
> > - This is an important point. We will implement and use both [4] and Jiang et al for estimating off-policy values, and show the results of both.  We will also update the discussion to be more frank about limitations here.
> > - Fair point, and we will revise our discussion and conclusions to be more critical about the limitations of RL in clinical settings, and how difficult it is to evaluate. The best way to see if the learned policies are actually useful is to try using them in practice, but barring that, we can conduct extensive clinical chart reviews to see if they recommend sensible treatments and if they are over-treating.

---

> ### Comment · AnonReviewer1 · 2018-01-14
> **Decision not to revise review**
>
> I appreciate the authors' in-depth and very thoughtful responses to all of the reviews. I really REALLY like this work, and contrary to the other (IMO, overly negative) reviews, I feel that it fits at ICLR, which has recent history of accepting very solid clinical application work, even without significant methods novelty.
>
> The reason I could not justify raising my score is that the response, while thoughtful, did little to help me understand the paper in a new light. Most of my critiques (and those in the other reviews) focused on exposition and discussion, but the authors did not provide a revised manuscript, so it's difficult to imagine how much the paper would be improved with some of the content written in the responses. I realize that making substantial revisions over the holidays is a drag (and that revising a manuscript during reviews is not normal practice in machine learning academia), but I typically do not feel comfortable assigning a higher score without seeing the revised manuscript.
>
> For what it's worth, I can imagine one circumstance where I'd make an exception: if the authors provided a new batch of experimental results on an open data seat like MIMIC.

---

### Public Comment · ~ZHUOWEI_WANG1 · 2017-12-14
**Very impressive work in the area of treating sepsis with DRQN and GP**

Dear author,
Thank you for you great work!!!
You almost include all the method I encountered in this area and I was so impressed by the mortality rate you have reduced  via the policy generated by your frame work. And I am very interested in your data preprocessing method and model.
Would you please release you source code for me to have a glimpse of you model?
And would you please give me more information of the specific data used in your article if that is  possible?
Lastly, could you please tell me what features did you choose in your POMDP?
Email: zhuowei.wang.cs.uts@gmail.com

---

### Public Comment · (anonymous) · 2017-12-16
**Public Source Code**

The idea is novel and the experiment results are the best among the papers in treating sepsis.
However, some details of the model design and experiment implementations need to be clarified with help of source code and the author mentioned source code will be released via Github.
So would it be okay that the author of this paper release the source code please?

---

### Public Comment · ~Lu_Liu4 · 2017-12-22
**Request for Code Please**

Dear Authors,

I am part of a team at University of Technology Sydney participating in the ICLR 2018 Reproducibility Challenge. We have chosen to reproduce your study and are wondering if you would like to share some or all of the code you used please!!

email: lu.liu-10@student.uts.edu.au

Thank you!

---

### Decision · Program_Chairs · 2018-01-29
**ICLR 2018 Conference Acceptance Decision**

**Decision:**

Reject

**Comment:**

This paper brings recent innovations in reinforcement learning to bear on a tremendously important application, treating sepsis.  The reviewers were all compelled by the application domain but thought that the technical innovation in the work was low.  While ICLR welcomes application papers, in this instance the reviewers felt that the technical contribution was not justified well enough.  Two of the reviewers asked for a more clear discussion of the underlying assumptions of the approach (i.e. offline policy evaluation and not missing at random).  Unfortunately, lack of significant revisions to the manuscript over the discussion period seem to have precluded changes to the reviewer scores.  Overall, this could be a strong submission to a conference that is more closely tied to the application domain.

Pros:
- Very compelling application that is well motivated
- Impressive (possibly impactful) results
- Thorough empirical comparison

Cons:
- Lack of technical innovation
- Questions about the underlying assumptions and choice of methodology